# Ozone Profile Retrieval Algorithm Based on GEOS-Chem Model in the Middle and Upper Atmosphere

Yuan An [1,2,3], Xianhua Wang [1,2,3,*], Hanhan Ye [1,3], Hailiang Shi [1,2,3], Shichao Wu [1,3], Chao Li [1,2,3] and Erchang Sun [1,2,3]

1   Anhui Institute of Optics and Fine Mechanics, Hefei Institutes of Physical Science, Chinese Academy of Sciences, Hefei 230031, China; aydr1994@mail.ustc.edu.cn (Y.A.); yehanhan@aiofm.ac.cn (H.Y.); hlshi@aiofm.ac.cn (H.S.); wusc@aiofm.ac.cn (S.W.); chaoli66@mail.ustc.edu.cn (C.L.); sunerchang@mail.ustc.edu.cn (E.S.)
2   Science Island Branch, Graduate School of University of Science and Technology of China, Hefei 230026, China
3   Key Laboratory of General Optical Calibration and Characterization Technology, Hefei Institutes of Physical Science, Chinese Academy of Sciences, Hefei 230031, China
*   Correspondence: xhwang@aiofm.ac.cn

**Abstract:** Ozone absorbs ultraviolet radiation, which has a significant impact on research in astrobiology and other fields in that investigate the middle and upper atmosphere. A retrieval algorithm for ozone profiles in the middle and upper atmosphere was developed using the spectral data from the TROPOspheric Monitoring Instrument (TROPOMI). A priori ozone profiles were constructed through the Goddard Earth Observing System-Chem (GEOS-Chem) model. These profiles were closer to the true atmosphere in the spatial and temporal dimensions when compared to the ozone climatology. The TpO3 ozone climatology was used as a reference to highlight the reliability of the a priori ozone profile from GEOS-Chem. The inversion results based on GEOS-Chem and TpO3 climatology were compared with ground-based ozone measurements and the satellite products of the Microwave Limb Sounder (MLS) and the Ozone Mapping and Profiles Suite_Limb Profile (OMPS_LP). The comparisons reveal that the correlation coefficient $R$ values for the inversion results based on GEOS-Chem were greater than 0.90 at most altitudes, making them better than the values based on TpO3 climatology. The differences in subcolumn concentration between the GEOS-Chem inversion results and the ground-based measurements were smaller than those between TpO3 climatology results and the ground-based measurements. The relative differences between the inversion results based on the GEOS-Chem and the satellite products was generally smaller than those between the inversion results based on TpO3 climatology and the satellite products. The mean relative difference between the GEOS-Chem inversion results and MLS is −9.10%, and OMPS_LP is 1.46%, while those based on TpO3 climatology is −14.51% and −4.70% from 20 to 45 km These results imply that using a priori ozone profiles generated through GEOS-Chem leads to more accurate inversion results.

**Keywords:** ozone-profile retrieval; GEOS-Chem; the middle and upper atmosphere





## 1. Introduction

Ozone was the most important trace gas in the atmosphere. Its absorption of ultraviolet radiation (UV) was intricately related to the search for extra-terrestrial life, microbial survival and reproduction, and the design of near-space vehicles that function in the middle and upper atmosphere [1–3]. It also plays a significant role in the dynamics, chemistry, and physics of the middle and upper atmosphere. The vertical distribution of ozone has a fundamental influence on the heating rate of the atmosphere and the formation of the inversion layer. Additionally, ozone can control the physical and chemical state of the middle and upper atmosphere directly and/or indirectly because of its strong oxidative properties and because it was a source of another important oxidant, OH. The study of ozone profiles with high spatial resolution, timeliness, and accuracy will better meet the

requirements of applications such as UV index forecast, changes in atmospheric circulation, and principles of atmospheric photochemical reactions.

Ozone profile monitoring requires the accurate measurement of its vertical distribution at high spatial and temporal resolution. An ozonesonde and ozone lidar were used for this purpose. Nevertheless, neither of these technologies can meet the demand for global coverage because of the limitations imposed by the sparse distribution of the stations. With the development of satellite technology, the use of satellite data has become a more viable option for obtaining global vertical distributions of ozone [4]. Satellite payloads such as the Odin Optical Spectrograph and InfraRed Imaging System (OSIRIS), SCanning Imaging Absorption spectrometer for Atmospheric Cartography (SCIAMACHY), Microwave Limb Sounder (MLS), and Ozone Mapping and Profiles Suite_Limb Profile (OMPS_LP) scan the atmosphere to generate an ozone profile product with high accuracy and vertical resolution in limb observation mode [5]. However, few limb payloads were available to provide observation data at present, and only a few new limb observation missions were planned for the future. Plans for a nadir observation mode have existed for years. The along-track spatial resolution of a nadir mode was much higher than that of the limb mode. Ozone has strong absorption characteristics in the Hartley (200–310 nm) and Huggins (310–350 nm) bands. Moreover, the transmission capacity of UV radiation was closely related to the wavelength and ozone concentration. These theoretical bases between ozone and UV radiation, together with Rayleigh scatter theory, were used to obtain ozone profiles from atmospheric backscattered UV radiation in nadir mode [6,7].

An a priori profile that was close to the true environment was a key factor in improving the retrieval accuracy in the retrieval algorithm [8–10]. The a priori ozone profile was a significant parameter in atmospheric ozone inversion. The ozone profiles were retrieved using up to 12 discrete bands of information, with a priori profiles retrieved from a database constructed from the data of the Stratospheric Aerosol and Gas Experiment (SAGE) and ozone lidar [11]. Advances in optical and aerospace technology allowed the Global Ozone Monitoring Experiment (GOME) to perform the first satellite-based measurement of continuous spectra in the UV and visible light spectra at high spectral resolution. Optimal estimation inversion technology proved able to retrieve ozone profiles in the troposphere and lower stratosphere when a suitable a priori ozone profile was used [12]. The ozone-profile retrieval algorithm was first developed based on the data from GOME [13]. The accuracy of inversion results was verified by ozone lidar when the data from absolute GOME calibration were first used [14]. A priori ozone profiles were all derived from ozone climatology constructed from ozone lidar and satellite data. Tropospheric ozone inversion results were further improved through spectral and wavelength corrections of GOME measurements. A priori ozone profiles were derived from the TOMS V8 ozone climatology [15]. The spatial resolution of measurement improved with the introduction of the Ozone Monitoring Instrument (OMI) and GOME-2. The Ozone-profile retrieval Algorithm (OPERA) was used to obtain a long time series of ozone profiles using the data from the GOME series. A priori ozone profiles were obtained from three different ozone climatology systems: Ozone Climatology by Fortuin & Kelder, TOMS Ozone Climatology, and Ozone Climatology by McPeters et al. [4]. An algorithm for the high-accuracy inversion of tropospheric ozone was developed based on GOME-2 measurements [16]. The accuracy of tropospheric ozone profile inversion was 10% after spectral recalibration of OMI Level 1 data. Both a priori ozone profiles were derived from the ozone climatology constructed by McPeters et al. [17]. The UV-Vis and SWIR spectral data were measured using the TROPOspheric Monitoring Instrument (TROPOMI) in nadir mode. Its unique advantage was the high-spatial resolution (28 km × 5.5 km) of the UV1 band, which was sensitive to ozone in the middle and upper atmosphere. The version 1 Level 1 data for band 3 were used to analyze the changes in the tropospheric ozone profile. A priori ozone profiles were derived from the ozone climatology by Bak et al. [18]. The Tikhonov regularized Ozone-profile retrieval with SCIATRAN (TOPAS) algorithm was used to retrieve the ozone profile from 0 to 60 km using the version 2 Level 1 data of band 1 and band 2 [19]. The

combination of TROPOMI UV and Cross-track Infrared Sounder (CrIS) IR data was used to improve the performance of ozone-profile retrieval in term of vertical resolution and accuracy in the troposphere and stratosphere. Both a priori ozone profiles were derived from the ozone climatology by Lamsal et al. [6]. In general, the priori profiles used in the ozone-profile retrieval algorithm were always derived from ozone climatology such as TMOS V8 climatology, climatology by Lamsal et al., climatology by McPeters et al., and others. These climatologies were obtained using statistical analysis of exiting multi-source observation data.

Although continuous updates have been made, the existing methods for designing the a priori ozone profiles have the problem of greatly deviating from the actual environment, which affects the accuracy of inversion. Ozone climatology based on the data from both payloads on the ground and onboard balloons and satellites is the main source of a priori ozone profiles. However, mathematical methods were required to deal with the problems arising from using different sources, times, and spatial scales when the climatology was constructed. The spatial and temporal variability characteristics of the ozone profile were smoothed. Although some auxiliary indicators such as the range of total column ozone or tropopause height were added to the climatology to improve the accuracy of a priori ozone profiles, it was still difficult to show changes in the ozone profile in the spatial and temporal dimensions. The atmospheric chemical transport model was becoming more mature and reliable with the development of a physical and chemical theory of the atmosphere, the use of computer technology, and other developments. It was possible to provide a more accurate a priori ozone profile using atmospheric chemical models [20]. The atmospheric chemistry transport model was driven by emission data and highly time-sensitive meteorological data. An a priori ozone profile can be modelled to match the needs of the retrieval algorithm in terms of both time and region. An a priori ozone profile based on the simulation results of GEOS-Chem has been evaluated. The feasibility of this approach was high when the data are used in the retrieval algorithm used to generate the tropospheric ozone profile for the Tropospheric Emissions: Monitoring of Pollution mission [21].

In summary, ozone profiles for the middle and upper atmosphere (from 15 to 60 km) were retrieved using optimal estimation technology applied to spectral data from TROPOMI Level 1, version 2. The a priori ozone profiles of the retrieval algorithm were based on the simulation results of GEOS-Chem. This was the first time that GEOS-Chem profiles have been used to generate a priori ozone profiles that can be used to retrieve ozone data from TROPOMI measurements. The Tropospheric Ultraviolet and Visible Radiation model (TUV) was used to calculate the relationship between the UV radiation and changes in ozone in order to illustrate the significance of applying high-accuracy ozone profiles from the middle and upper atmosphere.

## 2. Data and Model

The GEOS-Chem model and data from TROPOMI, MLS, OMPS_LP, ozonesonde, and ozone lidar were used in this study. The TROPOMI-measured radiances were used to retrieve the ozone profile. The GEOS-Chem model was used to construct the a priori ozone profile in the retrieval algorithm. The data from ozonesonde, ozone lidar, MLS, and OMPS_LP were used to validate the inversion results.

### 2.1. MLS

MLS is a microwave sounder in limb mode that obtains ozone profile information from a radiation of 240 GHz. It is on board Aura that works in a sun-synchronous orbit with an equatorial crossing time of 13:45 LT. The measurements obtained from the MLS and those obtained from TROPOMI are close in value, with a maximum distance of 1000 km and a time difference of 1.5 h [19]. Therefore, the Level 2 MLS ozone profile products were suitable for validating the inversion results in the spatial and temporal dimensions.

Version 5.0 products were used because this version provides more accurate forward model simulation when compared to previous versions. The effective altitude range of these products is from 0.05 to 68 hPa. The precision is 2–7% at 0.46–68 hPa and 20–50% at 0.05–0.46 hPa. The accuracy is from 5% to 15% at 0.05–68 hPa [22]. The MLS has been extensively used in the validation of other ozone profile results because of its temporal stability and accuracy [6,19,23–27].

Equation (1) was used to convert pressure to kilometers in altitude [28], as follows:

$$Z_i = -7ln(P_i/p_0), \ p_0 = 1000 \ \text{hPa} \tag{1}$$

where $P_i$ is the pressure of the MLS product. Moreover, the MLS ozone profile is in units of volume mixing ratio ($\times 10^{-6}$). Equation (2) was used to convert it to number density [29], as follows:

$$ND_i = 7.244 * 10^{10} * VMR_i * P_i/T_i \tag{2}$$

where $ND_i$ is the number density (molecules/cm$^3$) at the $i$th layer; $VMR_i$ is the volume mixing ratio ($\times 10^{-6}$) at the $i$th layer; $T_i$ is the temperature in $K$ at the $i$th layer, which was obtained from the same version as the MLS temperature product.

*2.2. OMPS_LP*

The ozone profile products of OMPS_LP were used for comparison with the inversion results. OMPS_LP is the only spectrometer in the OMPS that obtains the solar radiance scattered by earth's atmosphere for the retrieval of ozone profiles through limb observation. It is located aboard the Suomi National Polar-orbiting Partnership (SNPP), which has the same orbit as Aura, and the ascending node was at 13:30 LT [25]. It operates at a close distance to Sentinel-5P (S5P). That means that comparisons between the products of OMPS_LP and the inversion results can be made similar conditions.

The OMPS_LP has three slits, expanding its cross-track coverage. Each slit corresponds to a 112 km vertical range at a tangent point through a 1.85° vertical field of view. These hardware designs enable OMPS_LP to cover an altitude from 0 to 60 km. The scattered solar radiance in the spectral range from 290 to 1000 nm was collected simultaneously by a charge-coupled device. The UV- and visible-spectrum data were used to obtain the ozone profiles for different altitude ranges of the atmosphere separately [27,30]. The version 2.6 ozone profile products were used in the study. The precision is in the range of 3–4% between 20 and 50 km. Some larger values for retrieval precision (10–20%) are generated in the lower stratosphere. Systematic errors from measurements produce a retrieval error of ±3% [31]. OMPS_LP has also been widely used for the validation of other ozone profile products [18,19,32,33].

*2.3. TROPOMI*

TROPOMI is the only payload on board S5P. It consists of four spectrometers, which measure radiation information in the UV, visible, near-infrared and shortwave infrared wavelength ranges. S5P is in a sun-synchronous orbit with an equator passing time of 13:30 LT.

Version 2 L1B data from band 1 (267–300 nm) and band 2 (300–332 nm) were used to retrieve ozone profiles for the middle and upper atmosphere. The data have been further recalibrated, which means that this version has sufficient quality to retrieve the ozone profiles, unlike the data from version 1 L1B [34]. The spectral resolution of both is 0.5 nm and the spectral sampling is 0.065 nm. The spatial resolution is 28 km × 5.5 km in band 1 and 3.5 km × 5.5 km in = band 2. The radiation of band 1 is affected by the Hartley absorption band. Therefore, it is sensitive to changes in ozone concentrations in the middle and upper atmosphere. The pixels had to be binned to obtain an adequate signal-to-noise ratio because of the much lower intensity of radiation in band 1. The pixels of band 1 and band 2 were matched and binned to allow for ozone-profile retrieval due to the differences in sampling. The five pixels of band 1 were binned in the along-track dimension, while the

eight pixels of band 2 matched to band 1 were binned in the across-track dimension. That approach yielded a spatial resolution of about 28 km × 28 km. The spectral calibration and an additional spectral correction called soft calibration were introduced following the methods proposed by Mettig et al. [19]. The pixels used in the retrieval algorithm met the requirements of some quality-control fields associated with the TROPOMI L1B product, including 'Measurement Quality', 'Spectral Channel Quality', and others.

### 2.4. GEOS-Chem

GEOS-Chem is an improved global three-dimensional atmospheric chemical-transport model that has been widely used in ozone-distribution studies. The simulated ozone results have been proven to have accuracy within the range of applications' requirements through comparison with data from satellites and ground-based and airborne instruments [35–37]. Classic GEOS-Chem v 14.1.1 was used to simulate the spatial and temporal distribution of an ozone profile, employing full chemistry type with standard simulation options. The photolysis mechanism was described by FAST-JX [38]. The stratosphere-troposphere exchange of ozone was taken from the linear ozone (Linoz) stratospheric ozone chemistry package [39]. The Kinetic PreProcessor was used to simulate the chemical kinetics [40]. The Harmonized Emissions Component (HEMCO) was used to compute atmospheric emissions from different sources, regions, and species on a user-defined grid [41]. The other parameters, such as time step of chemistry, emission, and radiation, were set to the typical values by the model. The meteorology data source was the Modern-Era Retrospective Analysis for Research and Applications (MERRA-2), which has a native resolution of 0.5° × 0.625° and 72 hybrid sigma/pressure levels. A subset of MERRA-2 was used in the GEOS-Chem model. The data are described in Table 1.

**Table 1.** The MERRA-2 used in the GEOS-Chem model.

| Filenames | Characterization Parameters | Time Resolution |
|---|---|---|
| MERRA2.yyyymmdd.A1.res.nc4 | Various surface field | 1 h |
| MERRA2.yyyymmdd.A3cld. res.nc4 | Cloud | 3 h |
| MERRA2.yyyymmdd.A3dyn. res.nc4 | Dynamic field | 3 h |
| MERRA2.yyyymmdd.A3mstC. res.nc4 | Precipitation and sublimation field | 3 h |
| MERRA2.yyyymmdd.A3mstE. res.nc4 | Precipitation and convection field | 3 h |
| MERRA2.yyyymmdd.I3. res.nc4 | Pressure, temperature, and humidity | 3 h |

### 2.5. Ozonesonde and Ozone Lidar

Ground-based measurements can provide precise data for in situ ozone profiles. The balloon-borne ozonesonde can obtain the ozone profile from the troposphere to the lower stratosphere. The measurement precision of this instrument is 3–5%, and the accuracy is 5–10% [42,43]. The ozone lidar can obtain the ozone profile in the upper stratosphere specifically. The estimated accuracy of data obtained by this instrument between 15 and 50 km is 5% [44]. The ozonesonde and ozone lidar data have been used to validate the data produced by satellites due to the superior performance of the former [6,18,19,45–48]. The locations of stations are shown in Figure 1, and basic information regarding them is listed in Table 2.

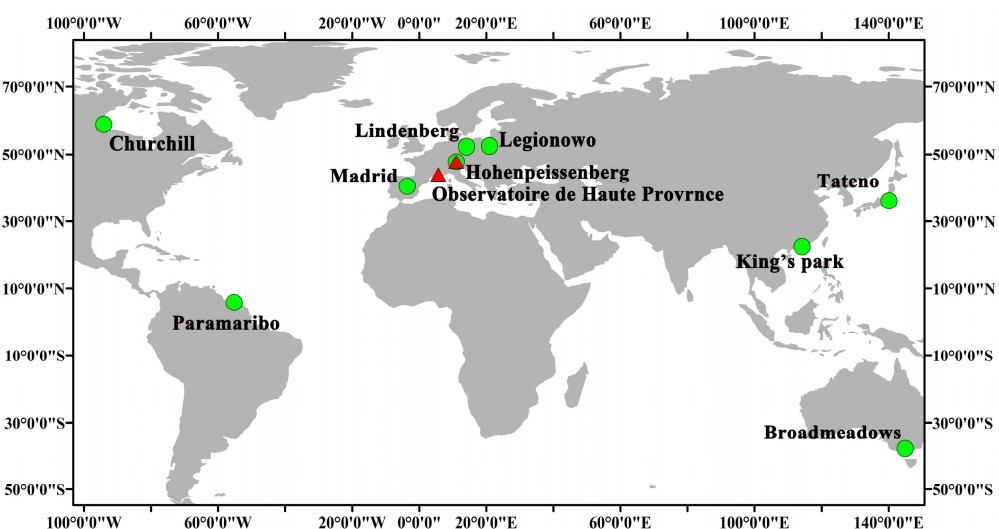

**Figure 1.** Distribution of ozonesondes and ozone lidar measurements used in this study.

**Table 2.** Ozonesonde and ozone lidar stations.

| Ozonesonde Station | Latitude, Longitude | Number of Profiles Used |
|---|---|---|
| Churchill | (53.30°N, 60.37°W) | 10 |
| Legionowo | (52.41°N, 20.96°E) | 12 |
| Hohenpeissenberg | (47.80°N, 11.07°E) | 12 |
| Lindenberg | (52.21°N, 14.12°E) | 12 |
| Madrid | (40.47°N, 3.58°W) | 12 |
| Tateno | (36.06°N, 140.13°E) | 11 |
| King's park | (22.31°N, 114.17°E) | 12 |
| Paramaribo | (5.81°N, 55.21°W) | 12 |
| Broadmeadows | (37.69°S, 144.95°E) | 12 |
| **Ozone Lidar station** | **Latitude, Longitude** | **Number of Profiles Used** |
| Hohenpeissenberg | (47.80°N, 11.07°E) | 47 |
| Observatoire de Haute Provence | (43.94°N, 5.71°E) | 46 |

The selection criteria for ground-based stations were that their data must be within 25 km maximum distance and 12 h maximum time of TROPOMI measurements. The ozonesonde data were from the World Ozone and Ultraviolet Radiation Data Centre. From all stations, nine were selected based on the temporal completeness of data and the changing characteristics of ozone profiles at different geographic locations. The selection of a greater number of European stations was based on the good continuity and comparability of their measurements, while the data from other stations were useful in characterizing the variability of ozone profiles in the different regions. Vaisala DigiCORA MW41 was used to obtain the atmospheric ozone concentrations with the processing software MW41 2.16.0. The meteorological variables were simultaneously measured by GPS radiosondes Vaisala RS41-SG. The ozone lidar data were obtained from the Network for the Detection of Atmospheric Composition Change. Two stations were selected to ensure the validity of ozone profiles throughout the year because the measurements were affected by weather conditions such as rainfall. The ozone lidar instrument measures data related to strong and weak absorption bands of ozone. The differential absorption inversion was used to determine all ozone profiles by using the difference between the atmospheric backscattered radiation and the ozone signal.

## 3. Retrieval Method

The radiation values from the atmosphere, rather than the target parameters themselves, were obtained by the sensor during the observation of remote sensing. The relationship between the state vector $x$, which represents the target parameters to be retrieved, and the observation vector $Y$ can be described by the forward model $F(x)$. It is shown in Equation (3), as follows:

$$Y = F(x) + \varphi \tag{3}$$

where $\varphi$ represents all errors. The solution of the state vector $x$ can be provided by the derivative Jacobian or weighting function $K$ (Equation (4)) of the forward model when the inversion problem is the linearization, as follows:

$$K = \frac{\partial F(x)}{\partial x} \tag{4}$$

The radiation values observed by the sensor were affected by multiple elements in the atmosphere. Therefore, it is an ill-posed problem to obtain the target parameters using radiation values. The derivative of the forward model could not be used to obtain the solution of the state vector $x$.

The retrieval algorithm of ozone profile in this paper was based on the principle of the optimal estimation method [49]. The difference between the simulated and measured radiances and between the a priori value and state vector $x$ can be minimized at the same time during the iterations [50,51]. The measurement error covariance matrix and the a priori error covariance were used to constrain the state vector $x$. The entire flowchart of the retrieval algorithm is shown in Figure 2.

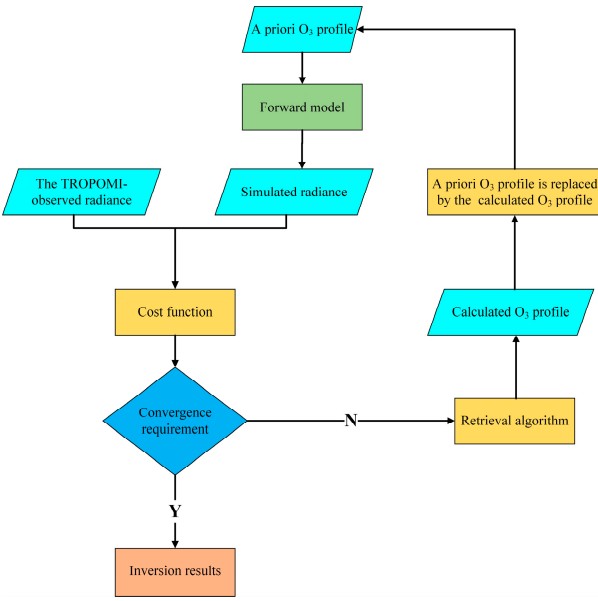

**Figure 2.** Flowchart of iterative inversion.

The inverse problem can be solved when the cost function $\chi^2$ that was given by Equation (5) takes the minimum value, as follows:

$$\chi^2 = (Y - F(x_m))^T S_\varepsilon^{-1} (Y - F(x_m)) + (x_m - x_a)^T S_a^{-1} (x_m - x_a) \tag{5}$$

where $x_m$ is the state vector at iteration step $m$; $F(x_m)$ is the simulation radiation computed by the forward model with the value $x_m$; and $S_a$ is the a priori error covariance matrix, which represents the natural variability of the ozone profile for each atmospheric layer. It can be obtained from the variability of long time-series ozone profiles in the GEOS-Chem

simulations [52–54]. $S_\varepsilon$ is the measurement error matrix, in which the diagonal elements were calculated through the SNR of TROPOMI. The Gauss-Newton iteration scheme was used as the iterative approach to this nonlinear problem. The solution is given at the m+1th iteration in the process of minimizing the residual using Equation (6), as follows:

$$x_{m+1} = x_m + \left(S_a^{-1} + K_m^T S_\varepsilon^{-1} K_m\right)^{-1} K_m^T S_\varepsilon^{-1} [Y - F(x_m) + K_m(x_m - x_a)] \tag{6}$$

The settings for the retrieval algorithm are shown in Table 3. The altitude range of the retrieval was from 15 to 60 km, with a vertical resolution of 1 km. The radiative transfer model SCIATRAN v4.5.5 was used as the forward model, assuming a pseudo-spherical atmosphere. The TROPOMI instrument response function V3.0.0 was used to convolve with the simulation spectra. Polarization and rotational Raman scattering were considered in the SCIATRAN. This process takes a long time, but the time cost was acceptable in scientific research as it would not have been in operational processes. The temperature and pressure profiles were taken from the ERA 5 reanalysis data from European Centre for Medium-Range Weather Forecasts. The cloud fraction was used to determine the pixels with and without clouds and was taken from the offline total ozone S5P product. The surface information was also taken from that product.

**Table 3.** Overview of settings for ozone-profile retrieval in the middle and upper atmosphere.

| Parameters | Setting |
|---|---|
| Radiative transfer model | SCIATRAN V 4.5.5 |
| Spectral characteristics | Spectral resolution: 0.5 nm; Spectral sampling: 0.065 nm |
| Spectral range | 270–329 nm |
| Altitude grid | 15–60 km, 1 km steps |
| Temperature and pressure | ECMWF ERA5 reanalysis data |
| Cloud fraction | Offline total ozone S5P product |
| Surface information | Offline total ozone S5P product |

It has been proven that accurate calibration of the spectral data and a near realistic simulation of the radiative transfer process were key to a successful retrieval process [17]. The initial setting of the a priori profile was one of the most critical parts of the simulation. The radiances measured using a nadir-viewing instrument do not contain sufficient vertical information to allow for the analysis of the ozone-absorption information needed to obtain the ozone profile. It was necessary to use suitable a priori profiles to compensate for the limitations of TROPOMI. Therefore, we carried out research to obtain and use a priori profiles with higher accuracy.

## 4. Construction of the a Priori Ozone Profile

The a priori ozone profile $x_a$ was a significant parameter in the retrieval algorithm. The appropriate settings for it were found based on the authors' knowledge of ozone and its distribution characteristics. Ozone is different from the stable atmospheric gases like carbon dioxide, nitrogen oxide, etc. The concentration of ozone is controlled by a series of chemical, radiative, and kinetic processes across a range of spatial and temporal scales. That means that ozone is generated by gas-phase photochemical reactions and is destroyed by reactions with chlorine, nitrogen, hydrogen, and bromine radicals [55]. The source and sink concentrations were also inconsistent across different regions and altitudes, while photochemical reaction rates depend on specific temperatures. This results in substantial variations in the spatial and temporal distributions of ozone profiles. Moreover, global climate change has an impact on ozone profiles. The concentrations of ozone have clearly decreased in the stratosphere because of the emission of halogenated ozone-depleting substances at the end of the twentieth century. However, the levels gradually increased again as the emission of these ozone-depleting substances was brought under control.

Brewer-Dobson circulation, which is associated with increased greenhouse gas concentrations, accelerated. That implies an increased transport of ozone to the middle and high latitudes. There are variances in the ozone levels at the bottom of the stratosphere across different latitudes. Thus, the altitude of peak ozone levels varies under the influence of atmospheric circulation. For example, the maximum ozone concentrations were observed at approximately 15 km or 20 km at high latitudes, while the peak altitudes were mostly around 30 km in lower latitudes [56]. In conclusion, the distribution of ozone profiles is complex and variable over time, as well as over the horizontal and vertical dimensions.

The a priori profiles used in the current ozone-profile retrieval with the optimal estimation method were almost always derived from ozone climatology. These climatologies were constructed using the ozone products from various satellites and other sensors. Several key factors were considered for use in improving the precision of ozone profile descriptions. Some climatologies take into account the complex nature of ozone's distribution across latitudes and altitudes and the asymmetry in the latitudinal profiles between the northern and southern hemispheres due to the variations in atmospheric temperature and circulation. The systemic profile changes resulting from seasonal variations in ozone photochemistry and atmospheric circulation were also considered. Currently, an a priori ozone profile can be precisely selected based on the latitude, month and total ozone column. However, it was difficult to construct an ideal ozone climatology that could reflect the interaction of multiple elements and was close to the real atmospheric environment. This difficulty arises from the dynamic variability of the ozone profile, which arises from photochemical reactions [57]. Constructing such a climatology, if feasible, would be a complex and extensive undertaking, making it less practical. In addition, the chosen a priori ozone profile did not match the actual measured conditions well in either space or time because of the fixed indices such as the range of latitude and time. It was also affected by the precision of the measurement of the total ozone column. These shortcomings affected the accuracy of the a priori ozone profile in describing the atmospheric conditions, resulting in suboptimal inversion accuracy.

Global atmospheric chemical transport models provide flexibility for user-defined conditions by incorporating updated meteorological data and well-estimated chemical mechanisms, along with information on emissions. GEOS-Chem has evolved into an effective tool for reproducing the spatial and temporal distribution of ozone profiles. A great number of stratospheric-related chemical reactions and coupled troposphere-stratosphere ozone reactions were included in GEOS-Chem. The stratospheric response was more comprehensively described on the basis of the original tropospheric chemistry reactions. The photochemical reactions of ozone in the middle and upper atmosphere were described by extending photolysis to the stratopause and calculating reaction rates at shorter wavelengths. The reanalyzed MERRA-2 meteorological data were produced by the GMAO/GEOS-5 Data Assimilation System. The inclusion of new observational data in data assimilation helped to improve the timeliness and accuracy of MERRA-2 data. The necessary emission data were supplied by the default settings of HEMCO. Hence, the simulated ozone profiles from GEOS-Chem served as a priori ozone profiles in the retrieval algorithm.

An excessively high spatial resolution in simulation was detrimental to accuracy and significantly increased the time cost because of the characteristics of the simulation and the limitations of the environment parameters [58]. A spatial resolution of $2° \times 2.5°$ was used, considering the requirements of a priori ozone profiles in the retrieval algorithm. The simulation results for the same day, as well as the weekly mean, half-monthly mean, and monthly mean were analyzed and compared with the ground-based ozone measurements to determine the reliability of each result in the temporal dimension. The comparisons of simulations under different time conditions are presented in Figure 3 for the Chinese station, with the data from the ozonesonde, and in Figure 4 for the German station, with the ozone lidar data.

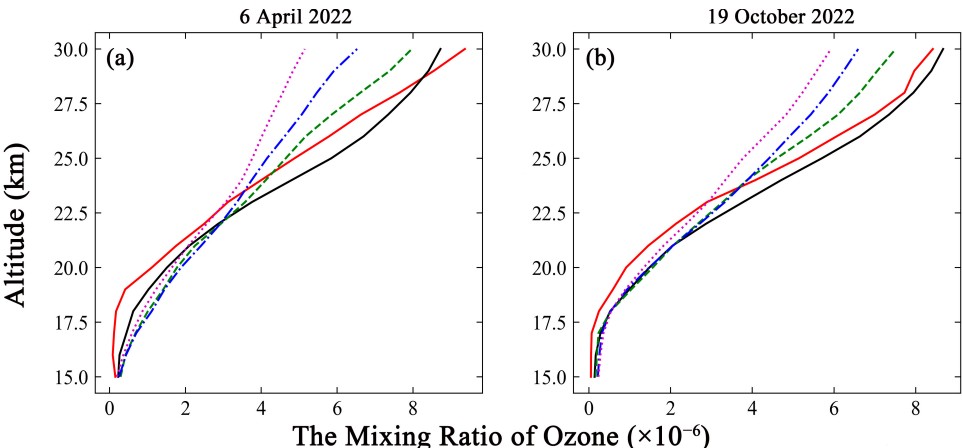

**Figure 3.** Comparison of ozonesonde measurements with a simulated ozone profile for different processing methods. The red solid line represents the ground-based measurements. The black solid line represents the simulation results for same-day measurements. The green dashed line represents the simulation results obtained by weekly mean methods. The blue dash-dotted line represents the simulation results obtained by half-monthly mean methods. The magenta dotted line represents simulation results obtained by the monthly mean method. (**a**) The comparison from 6 April 2022. (**b**) The comparison from 19 October 2022.

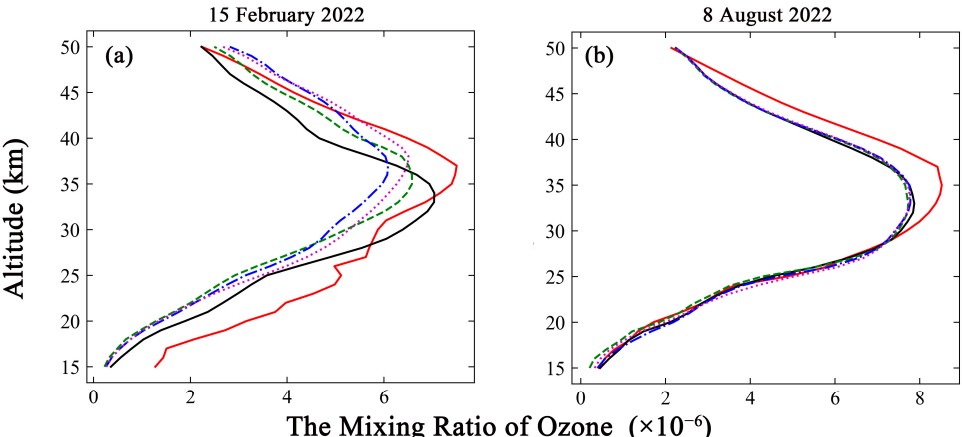

**Figure 4.** Same as Figure 3 but for comparison of ozone lidar measurements with simulated ozone profile for different processing methods. The red solid line represents the ground-based measurements. The black solid line represents the simulation results for same-day measurements. The green dashed line represents the simulation results obtained by weekly mean methods. The blue dash-dotted line represents the simulation results obtained by half-monthly mean methods. The magenta dotted line represents simulation results obtained by the monthly mean method. (**a**) The comparison at the 15 February 2022. (**b**) The comparison at the 8 August 2022.

The simulation results for the same day show good agreement with the ozonesonde measurements at the overlapping altitude range on both dates, as depicted in Figure 3. The $R$ values were 0.98 and 0.99, respectively. The weekly, half-monthly and monthly simulations on the two dates significantly differ from the ozonesonde measurements, with the variations ranging from 4.75% to 45.05% at altitudes above 25 km. The comparisons with ozone lidar measurements collected under the same conditions as the ozonesonde measurements are depicted in Figure 4. The $R$ values between the same-day simulations and lidar data were 0.89 and 0.98 on the two dates. The weekly, half-monthly, and monthly simulations on the two dates were different from the ozone lidar measurements, with differences between −42.99% and 25.85% at altitudes above 25 km. Accordingly, the distributions of the ozone profile were well reproduced by the GEOS-Chem simulation

results obtained by the same-day method. This approach can yield acceptable simulation accuracy and reduced time costs.

In summary, the ozone profiles were simulated on the target day with a horizontal resolution of $2° \times 2.5°$ through GEOS-Chem. The simulation results were used to construct a priori ozone profiles in the ozone-profile retrieval algorithm in the middle and upper atmosphere. However, the vertical resolution of the simulated ozone does not match the needs of the retrieval algorithm. The cubic spline interpolation algorithm was used to solve this issue. This algorithm ensures convergence and is characterized by a higher stability than other interpolation methods. It also maintains the continuity and smoothness of the interpolation function [59].

## 5. Results and Validations

The construction of an a priori ozone profile from TpO3 climatology was undertaken to conduct parallel inversion experiments, aiming to evaluate the reliability of the a priori ozone profiles derived from GEOS-Chem. TpO3 climatology has better vertical spatial coverage than other ozone climatologies. It can provide valid ozone profile data from the surface to 60 km. Secondly, TpO3 climatology includes comprehensive ozone profile information. Each month, it provides the ozone profiles for 18 latitudinal bands of 10 degrees. The percentages of ozone variability are given on a 1 km pressure altitude grid. Finally, the tropopause height is a novel index used in TpO3 climatology to classify the ozone profile, with the standard being latitude-month categorization. The tropopause height frequency was also provided. These features allow a more accurate characterization of the ozone profile [60].

TROPOMI L1B data were used to retrieve the ozone profile based on the retrieval method. A priori ozone profiles were obtained from GEOS-Chem and TpO3 climatology. The inversion results were validated through the ozonesonde and ozone lidar measurements. The Level 2 products of MLS and OMPS_LP were also used. The difference in vertical resolution affects the comparison results. The averaging kernels were always used to eliminate the effects when the inversion results were compared with the data of higher vertical resolution [19,49]. The data for comparison were first interpolated into the inversion results vertical grid. Then, they convolved with the averaging kernels of the inversion results.

### 5.1. Validations with the Ozonesonde and Ozone Lidar Measurements

The data from the ozonesonde and ozone lidar stations mentioned in Section 2.5 were compared with the two types of inversion results obtained from GEOS-Chem and TpO3 climatology. The data from the Legionowo ozonesonde station in Poland and the Observatoire de Haute Provence ozone lidar station in France were used as examples due to the large number of ozone profiles and the similarity of their results.

The direct comparison between the ozonesonde data that were closest to the TROPOMI observation pixel and the inversion results from different a priori ozone profiles are shown in Figure 5.

Figure 5 shows that the inversion results obtained from two types of a priori ozone profile were compared with the ozonesonde measurements at four time points. The vertical distribution trend of the ozone profile was consistent. The inversion results gradually increased with the atmospheric altitude, and the concentrations decreased with increased altitude after reaching a maximum. The minimum ozone concentration were found at the end of the retrieval altitude. The ozone concentrations were between 1.0 and $6.0 \times 10^{12}$ molecules/cm$^3$. However, the altitudes of the point of maximum concentration were not uniform. The altitude of peak ozone concentration was 19 km, as measured by the ozonesonde in spring. The corresponding altitudes found by GEOS-Chem and TpO3 climatology were both 20 km. The altitudes of peak ozone concentration in the GEOS-Chem inversion results were consistent with those from the ozonesonde measurements in summer and winter. They were 24 km and 21 km, respectively. In contrast, the altitude

of peak ozone concentration in the TpO3 climatology inversion results was 22 km. The altitude of peak ozone concentration in the ozonesonde measurements was similar to that in the GEOS-Chem inversion results in autumn, at 24 km and 23 km, respectively, while the corresponding value based on the TpO3 climatology inversion results was 21 km. There were spatial averaging effects in both GEOS-Chem and TpO3 climatology, and the ozonesonde data, being in situ measurements, provide a more accurate reflection of the ozone distribution in the true atmospheric environment. Therefore, the 1~3 km altitude difference between the inversion results and the ground-based measurements fell within the acceptable range. It was also seen that the inversion results based on GEOS-Chem were in closer agreement with the ozonesonde measurements in general.

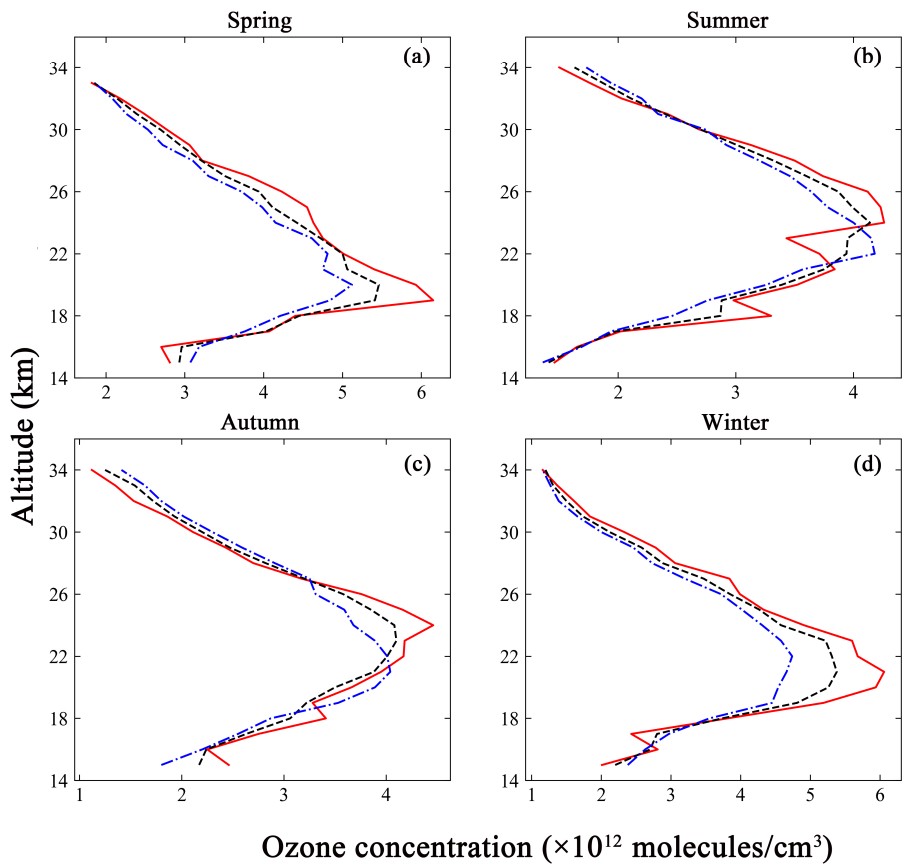

**Figure 5.** Comparison of the ozonesonde measurements (red solid line) with the inversion results based on different a priori ozone profiles. The black dashed line represents the inversion results based on GEOS-Chem. The blue dotted line represents the inversion results based on TpO3 climatology. (**a**) The comparison in spring. (**b**) The comparison in summer. (**c**) The comparison in autumn. (**d**) The comparison in winter.

Table 4 presents the ranges of relative difference $Difference_i$ between ozonesonde measurements and inversion results based on GEOS-Chem and TpO3 climatology. The relative difference $Difference_i$ was defined by Equation (7), as follows:

$$Difference_i = \frac{inv\_GC(or\ inv\_TpO3)_i - ground\_mea_i}{ground\_mea_i} * 100\% \tag{7}$$

where the $inv\_GC(orinv\_TpO3)_i$ represents the inversion results based on GEOS-Chem or TpO3 climatology at the $i$th altitude; $ground\_mea_i$ represents the ozonesonde measurements at the $i$th altitude.

**Table 4.** The range of $Difference_i$ between the inversion results and ozonesonde measurements.

| Season | Differences between GEOS-Chem Inversion Results and Ozonesonde Measurements | Differences between TpO3 Inversion Results and Ozonesonde Measurements |
|---|---|---|
| Spring | −11.90–9.73% | −21.23–17.99% |
| Summer | −12.97–14.98% | −25.57–20.81% |
| Autumn | −10.11–14.18% | −26.93–25.46% |
| Winter | −11.34–14.94% | −23.52–18.44% |

From Table 4, the range of $Difference_i$ between the inversion results based on GEOS-Chem and ozonesonde measurements was smaller than that between the results based on TpO3 climatology and ozonesonde measurements. The deviation of extremes of $Difference_i$ was also smaller for the former than for TpO3 climatology. That means that the GEOS-Chem inversion results were closer to the ozonesonde measurements and that their accuracy was higher than that of the TpO3 inversion results. The $R$ values from the scatterplots, lines of best fit, and subcolumn concentrations were more reliable due to the amount of data. The $R$ values at different altitudes were used to discuss the difference in order to display the comparison results more appropriately and avoid the influence of disturbance factors. The subcolumn concentrations were used to analyze the characteristics of values in different altitude ranges.

Figure 6 illustrates the comparisons and fitting of inversion results of different a priori ozone profiles and ozonesonde measurements at different altitudes using scatterplots and best-fir lines. The value of $R$ was also computed.

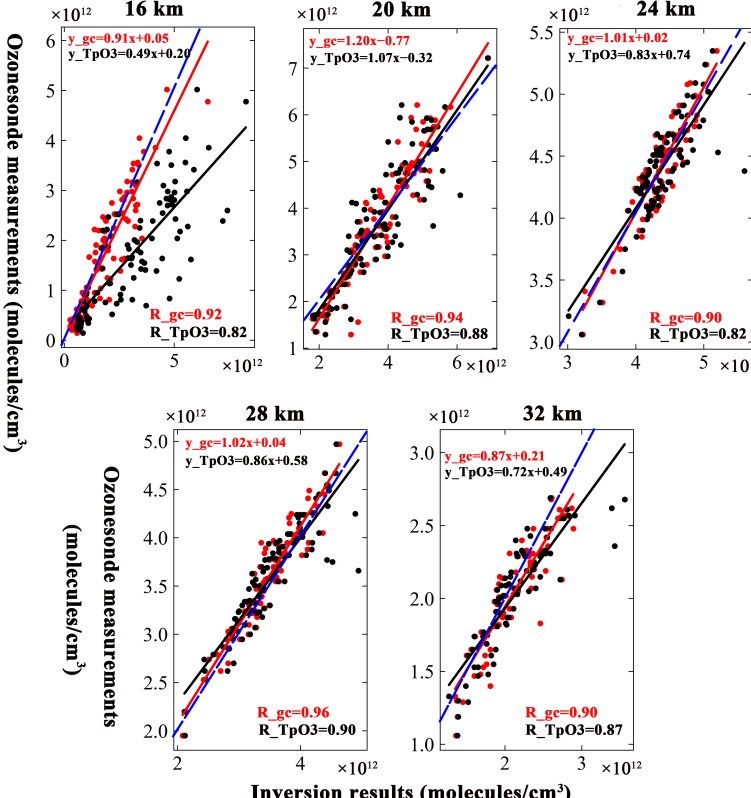

**Figure 6.** Scatterplots of inversion results and ozonesonde measurements. The solid lines show the linear fit. The $R$ values are shown in the top left corner of each picture. The comparison between the ozonesonde measurements and inversion results based on GEOS-Chem is shown in red, and the corresponding comparison of TpO3 climatology is shown in black. The 1:1 curve was plotted as a blue dashed line.

The scatter trend between the inversion results based on the GEOS-Chem and measured data was narrower compared to the trend based on TpO3 climatology at a 16 km altitude. The *R* also means that the GEOS-Chem inversion results have a better correlation with the measured data than do the results of TpO3 climatology. Although the scatter distribution between the inversion results and ozonesonde measurements was narrow at a 20 km altitude, the value of *R* for the inversion results based on the GEOS-Chem was greater than that for the inversion results based on TpO3 climatology. That means that the GEOS-Chem inversion results were closer to the ozonesonde measurements. The scatters between the inversion results and the ozonesonde measurements are wider at 24 km than at lower altitudes. This altitude was near the altitude of peak ozone concentration, and the variability in ozone concentration across different times and locations was more strongly reflected. The a priori ozone profile from GEOS-Chem better represents the adaptation to the changes. The GEOS-Chem inversion results show a better agreement with the measurements than the inversion results based on TpO3 climatology. The *R* value was above 0.90. The scatter trend becomes narrower again at a 28 km altitude. However, the *R* value between the inversion results based on GEOS-Chem and ozonesonde measurements were still larger than that between the inversion results based on TpO3 climatology and ozonesonde measurements. At a 32 km altitude, the scatter was wider than at 28 km. The degree of dispersion of scatter between based on GEOS-Chem and the ozonesonde measurements was smaller than that between the inversion results based on TpO3 climatology and the ozonesonde measurements. The *R* values were smaller than those at 28 km, but the *R* values for the correlation between the GEOS-Chem and TpO3 climatology data remained the same.

The inversion results based on different a priori ozone profiles show different error characteristics at each altitude. It was difficult to perform detailed analyses of every scenario. In addition to the scatterplots and *R* values analyzed above, the subcolumn concentrations were selected for comparison in two overlapping altitude ranges due to the limited range of altitudes available for ozonesonde measurements. The altitudes that could be effectively compared between the inversion results and ozonesonde measurements were approximately those from 16 to 32 km. The subcolumn at 16–24 km represents the lower altitude range, and the subcolumn at 24–32 km represents the higher altitude range. The subcolumn concentration was given by Equation (8) [29], as follows:

$$SCD = \sum ND_i * \Delta h * 3.7197e - 12 \tag{8}$$

where $SCD$ is the column concentration in Dobson Unit (DU) and $ND_i$ is the inversion result in number density (molecules/cm$^3$). The altitude interval $\Delta h$ is given in kilometers. The number density was converted to the column density at each *i*th layer and the subcolumn was then obtained by combining column densities at every layer. Figure 7 shows the differences in value between the subcolumns of the ozonesonde and the inversion results based on GEOS-Chem and TpO3 climatology in a scatterplot. Figure 7a shows the differences at 16–24 km, and Figure 7b shows the differences at 24–32 km.

The values of the differences between the subcolumn of the ozonesonde data and the subcolumn of the inversion results based on GEOS-Chem vary from −20.73 to 21.55 DU, with a standard deviation of 9.25 DU at 16–24 km (see Figure 7a), and from −17.41 to 11.24 DU, with a standard deviation of 6.37 DU at 24–32 km (see Figure 7b). The range of differences from −6.73 to 0.27 DU has a maximum probability of 26.73% in the lower altitude layer, and that from −7.01 to −1.81 DU has a maximum probability of 25.74% in the higher altitude layer. For TpO3 climatology, the values of the difference range from −27.60 to 25.65 DU, with a standard deviation of 11.80 DU, and from −19.57 to 16.15 DU, with a standard deviation of 7.78 DU. The range of differences from −0.90 to 8.00 DU has a maximum probability of 29.70% in the lower altitude layer, and that from −6.77 to −0.37 DU has a maximum probability of 21.78% in the higher altitude layer. The values of the differences between the subcolumns obtained from the inversion results based on GEOS-Chem and the subcolumns of the ozonesonde measurements were smaller than the

differences between the values based on TpO3 climatology and the subcolumns of the ozonesonde measurements.

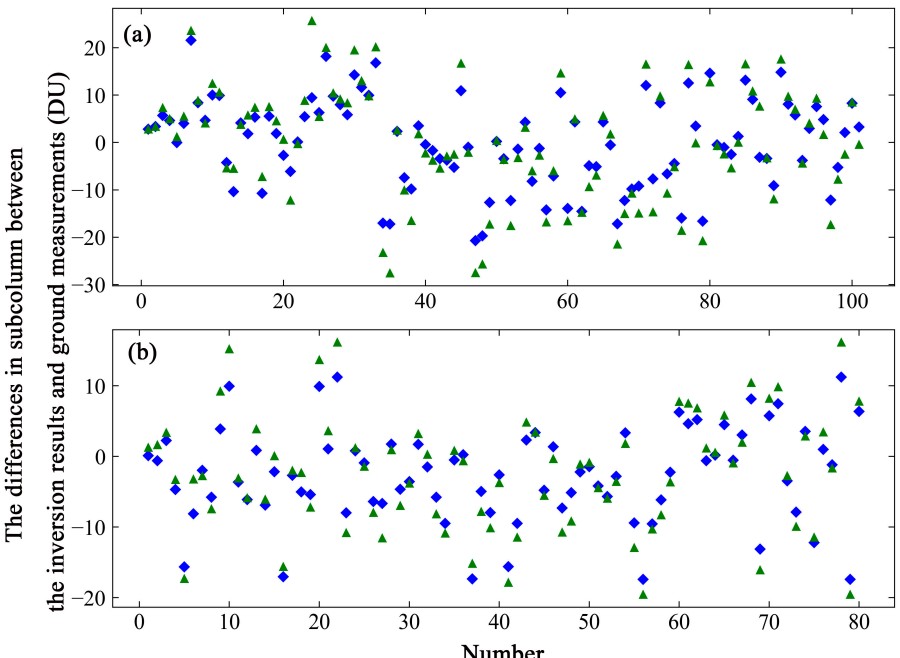

**Figure 7.** The scatterplot shows the differences in value in the subcolumns between the inversion results based on different a priori ozone profiles and the ozonesonde measurements. The green triangles represent the values of the inversion results based on TpO3 climatology, and the blue rhombuses those based on the GEOS-Chem data. (**a**) The values of the differences in the subcolumn at 16–24 km. (**b**) The values of the differences in the subcolumn at 24–32 km.

The direct comparison between the ozone lidar measurements, which were the closest in value to the TROPOMI observation pixels, and the inversion results from different a priori ozone profiles, is shown in Figure 8.

Figure 8 shows the comparison between the inversion results and the ozone lidar measurements at four time points. The vertical distribution trend of ozone profiles was the same as that for the ozonesonde. The ozone concentrations were between 0.9 and $6.0 \times 10^{12}$ molecules/cm$^3$. The altitude of peak ozone concentration was 20 km, as measured by the ozone lidar in spring. The altitude of peak inversion results based on GEOS-Chem was 21 km, and that for those based on TpO3 climatology was 22 km. The altitudes associated with the peak GEOS-Chem inversion values were consistent with the ozone lidar measurements in the other seasons. They were 25 km in summer, 23 km in autumn, and 22 km in winter. In contrast, the altitudes associated with the peak TpO3 inversion values were 23 km, 25 km, and 24 km, respectively. The difference in the altitudes associated with peak values between the ozone lidar measurements and the inversion results based on GEOS-Chem was smaller than the difference between the altitudes associated with peak values for the ozone lidar measurements and those associated with peak inversion results based on TpO3 climatology in all seasons.

Table 5 shows the ranges of relative difference $Difference_i$ between the ozone lidar measurements and inversion results based on GEOS-Chem and TpO3 climatology. The range and the deviation of extremes of $Difference_i$ between the inversion results based on GEOS-Chem and the ozone lidar measurements were also smaller than those between the inversion results based on TpO3 climatology and the ozone lidar measurements. The same results were found with the ozonesonde. The *R* values and subcolumn concentrations were also used to compare the inversion results with the ozone lidar measurements, as the same comparison reason as ozonesonde data.

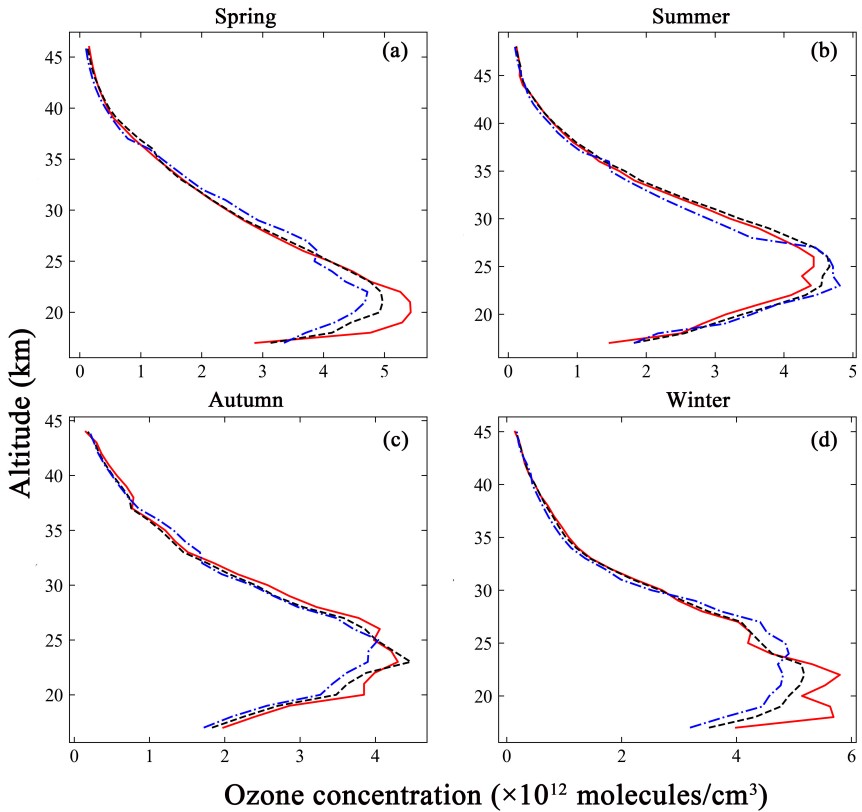

**Figure 8.** Similar to Figure 5 but showing the comparison of the inversion results with ozone lidar measurements (red solid line). The black dashed line represents the inversion results based on GEOS-Chem. The blue dotted line represents the inversion results based on TpO3 climatology. (**a**) The comparison in spring. (**b**) The comparison in summer. (**c**) The comparison in autumn. (**d**) The comparison in winter.

**Table 5.** The range of $Difference_i$ between inversion results and ozone lidar measurements.

| Season | Differences between GEOS-Chem Inversion Results and Ozone Lidar Measurements | Differences between TpO3 Inversion Results and Ozone Lidar Measurements |
|---|---|---|
| Spring | −19.92–10.21% | −36.46–16.12% |
| Summer | −4.20–23.75% | −18.87–24.47% |
| Autumn | −9.89–18.95% | −15.04–34.51% |
| Winter | −24.07–15.00% | −33.42–23.99% |

Figure 9 illustrates the same comparisons and fitting of inversion results, but for ozone lidar measurements at different altitudes.

The subcolumn concentrations were selected for comparison in three overlapping altitude ranges. The inversion results and ozone lidar measurements ranges could be effectively compared at altitudes from approximately 19 to 44 km. The subcolumn concentration at 20–28 km represents the low concentration range, and that at 28–36 km represents the middle concentration range. The altitude range of the highest concentration was defined as 36–44 km. Figure 10 shows the same comparison, but with ozone lidar measurements. Figure 10a displays the differences in the values of the subcolumn at 20–28 km; Figure 10b shows the values at 28–36 km; Figure 10c shows the values at 36–44 km.

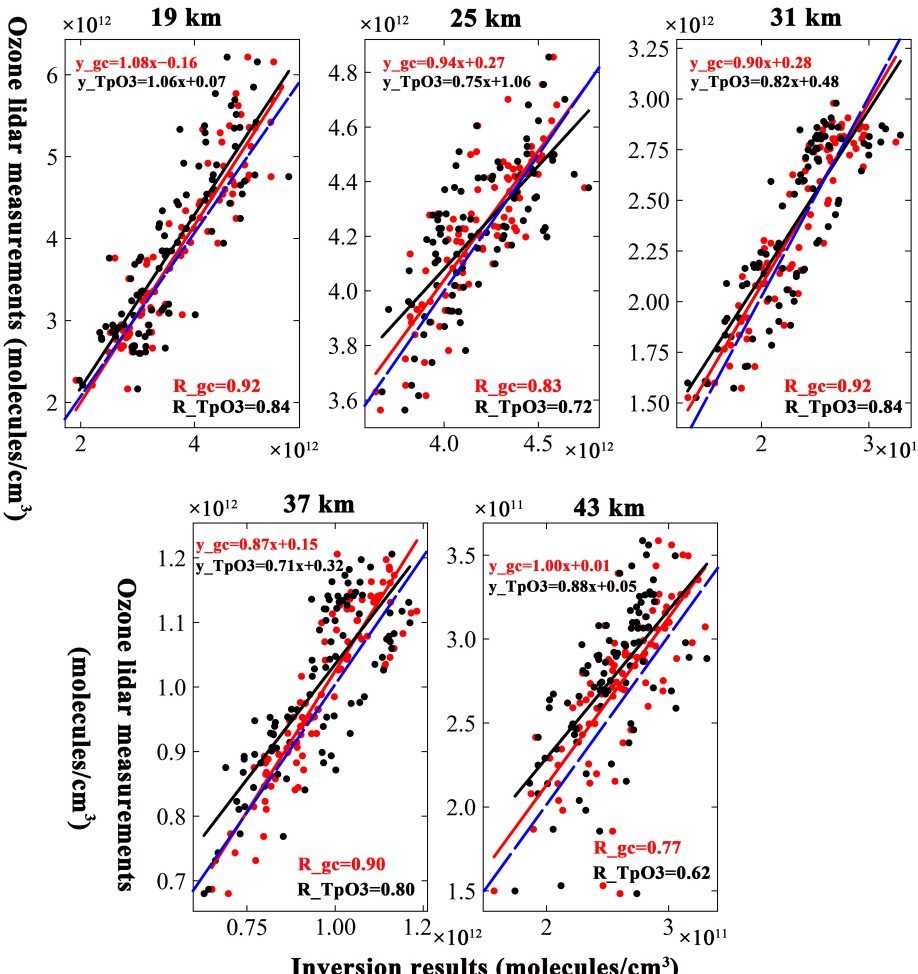

**Figure 9.** Same as Figure 6 but for ozone lidar measurements instead of ozonesonde measurements. The comparison between the ozonesonde measurements and inversion results based on GEOS-Chem is shown in red, and the corresponding comparison of TpO3 climatology is shown in black. The 1:1 curve was plotted as a blue dashed line. From Figure 9, it can be seen that the ozone lidar measurements were in better agreement with the inversion results based on GEOS-Chem than were the inversion results based on TpO3 climatology. The *R* values for the correlation between the GEOS-Chem inversion results and the ozone lidar measured data were greater than 0.90 between 19 and 37 km, and the corresponding *R* values of TpO3 climatology were greater than 0.79. The values at an altitude of 25 km were an exception. The spread scatters between the inversion results and ozone lidar measurements increased more at 25 km than at other altitudes. The *R* values were 0.83 for the correlation between the inversion results based on GEOS-Chem and the measured data and 0.72 for the correlation between the inversion results based on TpO3 climatology and the measured data. Even so, this was similar to the situation in which the inversion results were compared with the ozonesonde data, i.e., GEOS-Chem exhibits better performance than TpO3 climatology. At an of altitude 43 km, the scatter was wider than at 37 km. The inversion results based on the GEOS-Chem have lower variability than those based on TpO3 climatology. The *R* values were smaller than those for other altitudes, but the GEOS-Chem values there were also larger than those of TpO3 climatology. It may be that the errors become larger due to the thinner atmosphere and weaker ground-based or satellite measurement signals at higher altitudes. Generally speaking, the inversion results based on GEOS-Chem were closer to the ozone lidar measurements than were those obtained with TpO3 climatology.

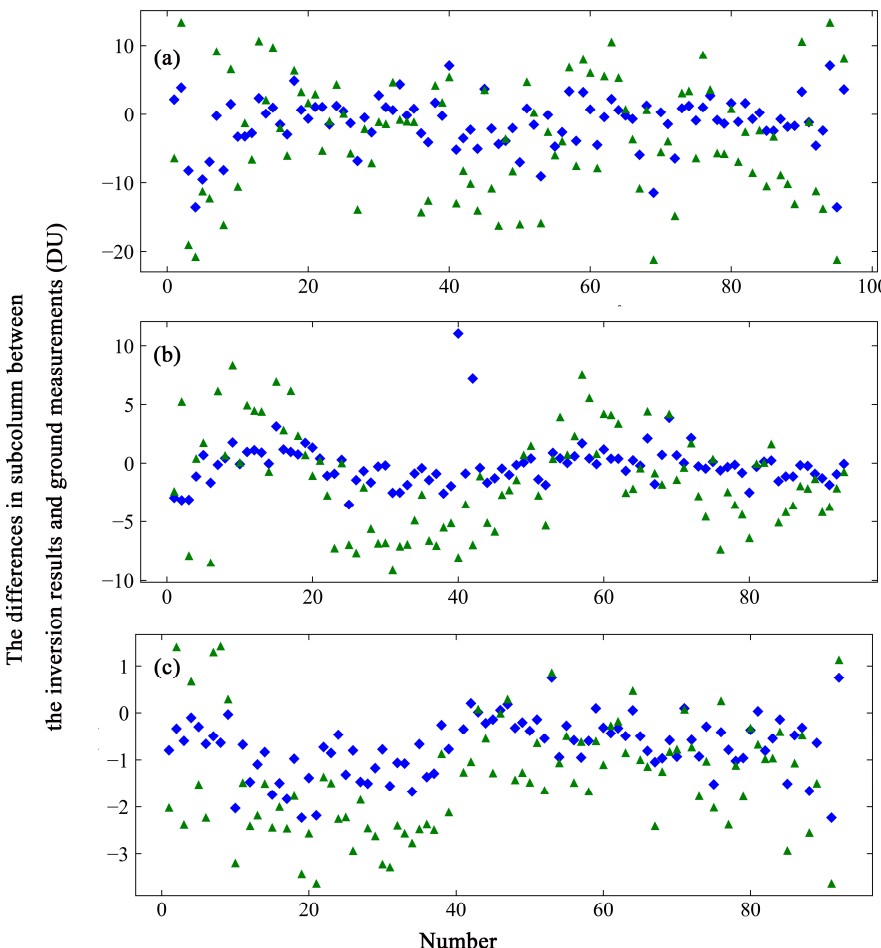

**Figure 10.** Same as Figure 7 but showing a comparison of the inversion results with the ozone lidar measurements. The green triangles represent the values of the inversion results based on TpO3 climatology, and the blue rhombuses those based on the GEOS-Chem data. (**a**) The values of the difference in the subcolumn at 20–28 km. (**b**) The values of the difference in the subcolumn at 28–36 km. (**c**) The values of the difference in the subcolumn at 36–44 km.

The values of the differences in subcolumn concentrations between the ozone lidar data and inversion results based on GEOS-Chem vary from −13.57 to 7.02 DU, with a standard deviation of 3.61 DU at 20–28 km (see Figure 10a), from −3.56 to 11.05 DU with a standard deviation of 1.94 DU at 28–36 km (see Figure 10b), and from −2.23 to 3.68 DU with a standard deviation of 0.74 DU at 36–44 km (see Figure 10c). The range of values of the difference from −3.27 to 0.16 DU has a maximum probability of 38.71% in the lower atmosphere layer; that from −0.64 to 0.82 DU has a maximum probability of 43.01% in the middle atmosphere layer; that from −1.05 to −0.46 DU has a maximum probability of 40.45% in the high atmosphere layer. The corresponding values of TpO3 climatology range from −21.24 to 13.25 DU with a standard deviation of 8.01 DU, from −9.15 to 8.32 DU with a standard deviation of 4.21 DU, and from −3.64 to 1.42 DU with a standard deviation of 1.13 DU. The probability distribution indicates that the range of values of the difference from −2.67 to 3.56 DU has a maximum probability of 27.96%, that from −3.91 to −2.16 DU has a maximum probability of 17.20%, and that from −2.63 to −2.12 DU has a maximum probability of 21.35%.

In summary, the inversion results from two types of a priori ozone profiles show nearly the same trend in the distribution of ozone concentrations. Ozone concentrations increase with the increase in altitude until the altitude of peak ozone concentration was reached. Then, they decrease up to the end of the altitude range. However, the altitudes of peak concentration varied. The GEOS-Chem inversion results were more consistent with

the ground-based measurements. Moreover, the comparison of *R* values and subcolumn concentrations between the inversion results based on GEOS-Chem and the ground-based measurements and between TpO3 climatology and the ground-based measurements at different altitude ranges indicates that the inversion results based on GEOS-Chem were more accurate and stable than those based on TpO3 climatology.

### 5.2. Validations with the Data of MLS and OMPS_LP

An ozone profile in the higher altitudes can be obtained by limb-viewing sensors. The inversion results based on GEOS-Chem and TpO3 climatology were also compared with the Level 2 products of MLS and OMPS_LP. The Chinese regions were taken as an example. The TROPOMI pixels were retrieved for a typical month in the different seasons and at a maximum distance of 50 km from the tangent point of both MLS and OMPS_LP. It was harder to match the spatial location between TROPOMI and MLS (or OMPS_LP) than between TROPOMI and the ground-based instrument because using a different observation mode leads to a difference in spatial resolution. It is thus difficult to obtain an accurate correspondence relationship between the data. Besides, fewer products of a limb sensor can be used compared to ground-based measurements under the rigorous selection parameters like quality and other criteria. Therefore, the inversion results in different latitude bands were compared to the limb products. The relative difference $Difference_{i-limb}$ (Equation (9)) was used to characterize the inversion accuracy due to the limited amount of data, as follows.

$$Difference_{i-limb} = \frac{inv\_GC \ (or \ inv\_TpO3)_i - Pro\_MLS \ (or \ Pro\_OMPS\_LP)_i}{Pro\_MLS \ (or \ Pro\_OMPS\_LP)_i} * 100\% \quad (9)$$

where $inv\_GC(or inv\_TpO3)_i$ represents the inversion results based on GEOS-Chem (or TpO3 climatology) at the *i*th layer. $Pro\_MLS(or Pro\_OMPS\_LP)_i$ was the Level 2 product of MLS (or OMPS_LP) at the *i*th layer.

The two types of inversion results were individually compared with the Level 2 products of MLS and OMPS_LP. Validations were performed for the overlapping altitudes due to the different effective altitudes of the Level 2 products. The relative deviations of limb $Difference_{i-limb}$ are shown in Figure 11.

As seen in Figure 11, (1) the trends of the relative deviations were almost the same between the inversion results and Level 2 products of MLS and OMPS_LP in spring. The comparisons between the inversion results and MLS reveal a large fluctuating trend in relative difference at lower altitudes. The amplitude of the oscillations tends to level off at altitudes from 20 to 40 km. The relative differences gradually increase with altitudes until 53 km. The comparison of inversion results and OMPS_LP indicates that violent oscillations appear from 16 to 21 km. The trends exhibit a staggered oscillation until the end of the altitude range. (2) In summer, the amplitude of oscillation trends in the relative difference between the inversion results based on GEOS-Chem and the MLS data was smaller than that between the inversion results based on TpO3 climatology and MLS data. That phenomenon was evident particularly at the bottom and top of the atmosphere. The same phenomenon also appears in the comparison between the two types of inversion results and OMPS_LP. (3) Each relative difference oscillates around −1.55% from 15 to 26 km and around about −12.56% from 42 to 60 km with the comparison to MLS data in autumn. The value increases monotonically at altitudes from 27 to 41 km. The amplitude of oscillation trends in the relative difference between the inversion results based on GEOS-Chem and TpO3 climatology and the MLS data in summer was consistent with the amplitude of the trends in the difference between the inversion results and the OMPS_LP data in autumn. (4) The relative differences between each set of inversion results and the MLS data show an interlocking oscillation in winter. Intense oscillations appear between 17 and 27 km. The same conditions appear when the inversion results are compared with the OMPS_LP data.

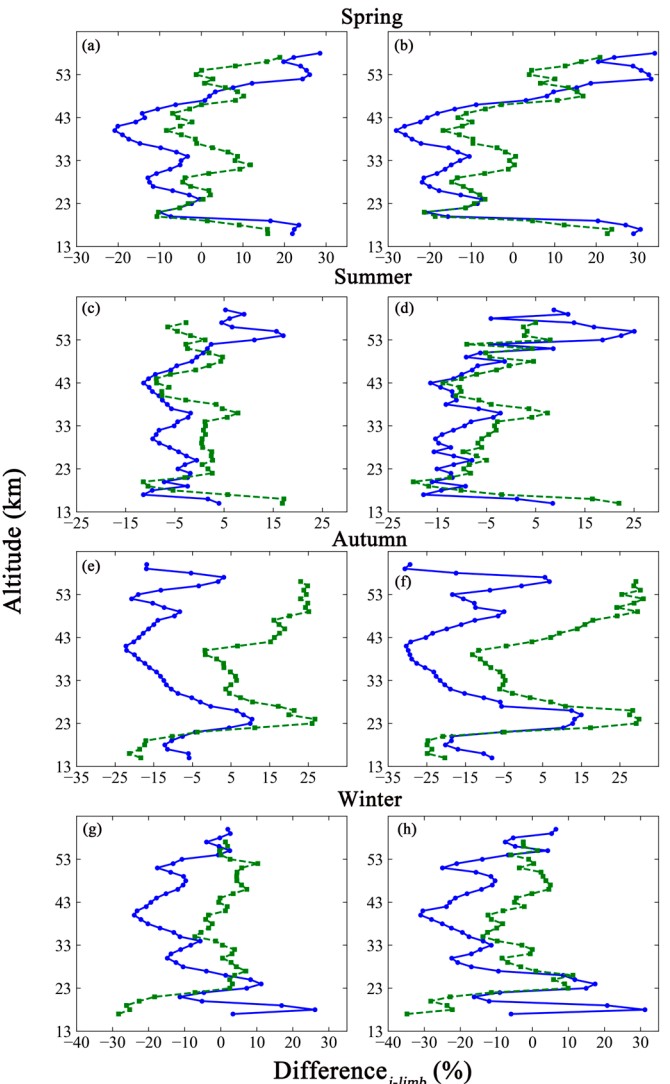

**Figure 11.** The relative difference in limb data between the inversion results based on GEOS-Chem (panel: (**a**,**c**,**e**,**g**)) and the Level 2 products of MLS (blue solid line with circle) or OMPS_LP (green dashed line with squares). The panels on the right (**b**,**d**,**f**,**h**) show the corresponding comparison with inversion results based on TpO3 climatology.

The statistics describing relative differences between each set of inversion results and the products of MLS and OMPS_LP were presented separately in Tables 6 and 7.

**Table 6.** Statistics describing the $Difference_{i-limb}$ between each set of inversion results and MLS products (unit: percent).

| Season | Maximum | | Minimum | | Means | | Standard Deviation | |
|---|---|---|---|---|---|---|---|---|
| | GEOS-Chem | TpO3 | GEOS-Chem | TpO3 | GEOS-Chem | TpO3 | GEOS-Chem | TpO3 |
| Spring | 28.52 | 34.04 | −20.83 | −28.11 | 0.18 | −2.42 | 15.25 | 21.03 |
| Summer | 17.04 | 25.04 | −11.43 | −17.85 | −2.42 | −5.41 | 6.91 | 11.18 |
| Autumn | 10.42 | 14.98 | −22.25 | −30.72 | −9.86 | −13.53 | 8.83 | 13.10 |
| Winter | 26.10 | 31.21 | −24.03 | −31.04 | −7.47 | −10.38 | 10.90 | 14.26 |

**Table 7.** Statistics describing the $Difference_{i-limb}$ between each set of inversion results and OMPS_LP products (unit: percent).

| Season | Maximum | | Minimum | | Means | | Standard Deviation | |
|---|---|---|---|---|---|---|---|---|
| | GEOS-Chem | TpO3 | GEOS-Chem | TpO3 | GEOS-Chem | TpO3 | GEOS-Chem | TpO3 |
| Spring | 18.89 | 23.76 | −10.69 | −21.35 | 2.25 | −1.18 | 7.46 | 12.22 |
| Summer | 17.15 | 21.93 | −11.43 | −19.90 | −0.13 | −3.57 | 6.05 | 8.22 |
| Autumn | 26.71 | 31.07 | −21.32 | −24.94 | 9.28 | 5.62 | 14.06 | 18.80 |
| Winter | 10.27 | 11.23 | −28.37 | −34.82 | −1.64 | −5.41 | 9.35 | 10.17 |

On average, the absolute value of relative differences between the GEOS-Chem inversion results and MLS was generally smaller that between the inversion results based on TpO3 climatology and MLS, as shown in Table 6. The relative differences between the inversion results based on GEOS-Chem and MLS have smaller oscillation amplitudes compared to those between the inversion results based on TpO3 climatology and MLS. (1) The relative difference in the GEOS-Chem inversion results ranges from −20.83% to 28.52%, which was smaller than those based on TpO3 climatology in spring. The maximum relative difference (34.04%) based on TpO3 climatology occurs at 58 km, the same altitude associated with the maximum value based on GEOS-Chem. These comparisons mean that GEOS-Chem can better reflect the distributions of ozone at low concentrations in the upper atmosphere. (2) The absolute values of the minimum and maximum relative difference of the GEOS-Chem inversion results were also smaller than those of the inversion results based on TpO3 climatology in summer. The maximum value of the GEOS-Chem inversion results occurred at 54 km, while that of the inversion results based on TpO3 climatology occurred at 55 km because of the low ozone concentrations. The minimum values both occurred at 17 km due to the poor quality of ozone radiation-absorption data. The dispersion of the relative difference of the inversion results based on GEOS-Chem was smaller than that of the inversion results based on TpO3 climatology. (3) The maximum relative differences of the inversion results occurred at 24 km and 25 km in autumn. The relative difference of the GEOS-Chem inversion results was smaller than that of the TpO3 inversion results. This result means that a priori ozone profiles from GEOS-Chem can better reflect the change in ozone concentration near the peak altitude. Moreover, the stability of the GEOS-Chem inversion results was also better than that of the inversion results based on TpO3 climatology. (4) The maximum and minimum relative differences of each set of inversion results both occurred at 18 km and 40 km in winter. However, their absolute values for the GEOS-Chem inversion results were 5.11% and 7.01% smaller than those for the inversion results based on TpO3 climatology. This result means that GEOS-Chem has better reference meanings in the middle and upper atmosphere. The dispersion of the relative differences of each set of inversion results was the same as in the other seasons.

Table 7 illustrates the statistics describing the values of $Difference_{i-limb}$ between each set of inversion results and OMPS_LP products. (1) The difference in extremes of $Difference_{i-limb}$ between the GEOS-Chem inversion results and OMPS_LP was 29.58% in spring, while the corresponding value for TpO3 climatology was 45.11%. The difference between GEOS-Chem inversion results and the OMPS_LP products was greater than that between the OMPS_LP products and the TpO3 inversion results due to the mean value. Generally, emission and/or the meteorological conditions may lead to such results in spring. However, the standard deviation of GEOS-Chem data was 4.76% smaller than that of TpO3 climatology data. Thus, the discrete nature of the GEOS-Chem inversion results resulted in better performance than was obtained from the inversion results based on TpO3 climatology. (2) In summer, the mean values show an underestimation of ozone profiles by both sets of inversion results. However, the difference in extremes of $Difference_{i-limb}$ was the smallest over the four scenarios. In summer, the corresponding value of the GEOS-Chem inversion results was 28.58%, and that of the TpO3 inversion results was 41.83%. The standard deviation exhibits the same trend as the difference in extremes. This result implies

that GEOS-Chem and TpO3 climatology have a better ability to reflect the atmospheric ozone profiles in summer than in other seasons. Nevertheless, GEOS-Chem was superior to TpO3 climatology. (3) In contrast, the difference in extremes of $Difference_{i-limb}$ was the greatest in autumn among the four scenarios. The corresponding value of the GEOS-Chem inversion results was 19.45% larger than that in summer, and the value of the inversion results from TpO3 climatology was 14.18% larger than that in summer. Those numbers may reflect an overestimation of ozone profiles by the inversion results, as shown by the mean values. (4) The difference in extremes of $Difference_{i-limb}$ between the GEOS-Chem inversion results and OMPS_LP was 38.64% in winter, while the corresponding value of TpO3 climatology was 46.05%. The mean value of GEOS-Chem was 3.77% smaller than that of TpO3 climatology. The standard deviation of GEOS-Chem was 0.82% smaller than that of TpO3 climatology. The dispersion level of the inversion results from GEOS-Chem and TpO3 climatology was almost identical throughout the four scenarios.

The relative difference $Difference_{i-limb}$ between the inversion results based on GEOS-Chem and the products of the two limb sensors was smaller than that between the inversion results based on TpO3 climatology and the two limb sensors. Besides, the amplitude of the oscillations between the inversion results based on GEOS-Chem and the products of two limb sensors was also smaller than that between the inversion results based on TpO3 climatology and two limb sensors. It can thus be seen that the accuracy and stability of the GEOS-Chem inversion results were better than those of the inversion results based on TpO3 climatology.

### 5.3. Effects on UV Radiation

The ozone concentrations in the middle and upper atmosphere change the amount of UV radiation, especially in the band from 280 to 315 nm (UVB). The values of change in UVB were evaluated here for this reason, using updated TUV radiative transfer model 5.3.1, which was developed by the National Center for Atmospheric Research. This model solves the multilayer atmospheric radiative transfer equation using a pseudo-spherical multi-stream discrete coordinate method. It also can calculate the UV irradiance from 280 to 320 nm and from 320 to 400 nm within the atmospheric altitude range of 0 to 120 km. It has been used in radiation calculation, atmospheric photochemical modelling, and ecological research because of its high accuracy [61,62].

The inversion results based on GEOS-Chem and TpO3 climatology were used to show the effect of different inversion accuracies on UVB radiative transfer processes. The percentage change $\Delta UVB_i$ at the $i$th layer indicates the level of UVB changes at some altitudes. It was calculated using Equation (10), as follows:

$$\Delta UVB_i = \frac{UVB_{i\_GC} - UVB_{i\_TpO3}}{UVB_{i\_TpO3}} * 100\% \tag{10}$$

where $UVB_{i\_GC}$ (or $UVB_{i\_TpO3}$) is the UVB irradiance that was affected by the inversion results based on GEOS-Chem (or TpO3 climatology) at the $i$th layer.

The UVB irradiance was strongly impacted by ozone concentration, as indicated by the statistics in Table 8. The UVB irradiance was minimally influenced by the inversion results based on different a priori ozone profiles in the upper stratosphere. However, the inversion accuracy of ozone profiles leads to a gradual increase in $\Delta UVB_i$ with the decrease in atmospheric altitudes. $\Delta UVB_i$ was 19.91% due to the variation in accuracy of the inversion results. Some inversion results cause $\Delta UVB_i$ to be greater than 50%. Research will be carried out on the stability and physical mechanism underlying such results in the future. UVB radiation has shorter wavelengths and higher energy, which leads to a strong photochemical effect. Thus, a change in the amount of UVB radiation has a significant impact on the ageing of long-time resident balloon materials, the survival and reproduction of microorganisms in the middle and upper atmosphere, and the global climate. The current focus was on improving the inversion accuracy of ozone column concentration,

and it was also important to improve the inversion accuracy of ozone profiles in the middle and upper atmosphere.

**Table 8.** The changes in UVB irradiance caused by inversion results based on different a priori profiles at different altitudes.

| Altitude (km) | $UVB_{i\_GC}$ (W/m$^2$) | $UVB_{i\_TpO3}$ (W/m$^2$) | $\Delta UVB_i$ |
|---|---|---|---|
| 55 | 16.34 | 16.36 | 0.12 |
| 45 | 15.31 | 15.52 | 1.37 |
| 35 | 10.54 | 11.26 | 6.83 |
| 25 | 5.38 | 6.09 | 13.13 |
| 15 | 2.90 | 3.48 | 19.91 |

## 6. Conclusions

The research needed to construct a retrieval algorithm for the ozone profiles in the middle and upper atmosphere was conducted in this paper. The TROPOMI version 2 Level 1 data of band 1 and band 2 were utilized, and the retrieval algorithm was developed based on the optimal estimation technique. One of the crucial factors in high-accuracy inversion concerns the a priori ozone profile settings, especially when using observation radiation data from the nadir-viewing sensors. GEOS-Chem has the ability to generate a priori ozone profiles. The research on the inversion accuracy of the ozone profiles was conducted based on the theory above and on the GEOS-Chem simulation results. A priori ozone profiles from TpO3 climatology were employed for comparison by conducting identical inversion experiments. The inversion results based on GEOS-Chem and TpO3 climatology were compared and validated against ozonesonde and ozone lidar measurements, along with Level 2 products of MLS and OMPS_LP, to establish their accuracy.

The superiority of the a priori ozone profile based on GEOS-Chem was verified through comparison with the data from the ozonesonde and ozone lidar. The mean *R* value of the correlation between the ozonesonde measurements and the GEOS-Chem inversion results was 0.92; however, it decreased to 0.86 when the ozonesonde measurements were compared with the TpO3 inversion results. The corresponding values for the ozone lidar measurements were 0.87 for correlation with the GEOS-Chem inversion results and 0.76 for correlation with the TpO3 inversion results. The correlation coefficient *R* values between the ground data and the inversion results based on GEOS-Chem were greater than that between the ground data and inversion results based on TpO3 climatology at every typical altitude. Moreover, the differences and standard deviations in subcolumn concentration between the ozonesonde (ozone lidar) measurements and the inversion results based on GEOS-Chem were smaller than those between the measurements and inversion results based on TpO3 climatology in every altitude range. That means the inversion results based on GEOS-Chem were more accurate than those obtained with TpO3 climatology. The relative differences between the GEOS-Chem inversion results and the MLS products ranged from −24.03% to 28.52%, and those between the GEOS-Chem inversion results and OMPS_LP products ranged from −28.37% to 26.71% across the four scenarios. The corresponding values for TpO3 climatology ranged from −31.04% to 34.04% and from −34.82% to 31.07%. The standard deviations of relative differences ranged from 6.91% to 15.25% in the comparison with MLS products and from 6.05% to 14.06% in the comparison with OMPS_LP products. The corresponding values for TpO3 climatology ranged from 11.18% to 21.03% and from 8.22% to 18.80%. The extreme values of relative difference always occurred at the bottom or top part of the atmospheric altitude for the inversion results because of the poor quality of the data on ozone radiation absorption and low ozone concentrations. However, the inversion results based on the GEOS-Chem model can better reflect the atmospheric ozone profiles than can those based on TpO3 climatology. Additionally, the relative differences between the GEOS-Chem inversion results and the products of two limb sensors were smaller in terms of the amplitude of oscillations than

were those between the inversion results based on TpO3 climatology and the two limb sensor products. The stability of the inversion results based on GEOS-Chem was also better than that of the inversion results based on TpO3 climatology. A priori ozone profiles based on the GEOS-Chem model can be more accurate than those based on TpO3 climatology.

Even though the improvements in ozone concentrations were not large in absolute terms, the changes in ultraviolet radiation were significant. This finding has significant implications for various applications, such as the survival of microorganisms and materials for near-space vehicles. The continuous development of GEOS-Chem, along with the continuous updating of related meteorological field data and chemical mechanisms, will further improve the accuracy and stability of ozone profiles based on it. This development will enhance the usability of profile inversion of the nadir-viewing observation data represented by TROPOMI.

**Author Contributions:** All authors contributed in a substantial way to the manuscript. Conceptualization, Y.A. and X.W.; methodology, Y.A., X.W. and H.Y.; validation, Y.A. and E.S.; investigation, C.L.; resources, X.W., H.Y. and H.S.; data curation, Y.A., S.W. and C.L.; writing—original draft preparation, Y.A.; writing—review and editing, Y.A. and X.W.; visualization, Y.A., E.S., S.W. and C.L.; supervision and formal analysis, S.W. and E.S.; project administration, X.W., H.Y. and H.S.; funding acquisition, X.W., H.Y., H.S. and S.W. All authors have read and agreed to the published version of the manuscript.

**Funding:** This research was supported in part by the National Key R&D Program of China (2022YFB3-901800 and 2022YFB3901804), and in part by the National Natural Science Foundation of China (NSFC) Young Scientist Fund (42205146).

**Data Availability Statement:** Data are contained within the article.

**Acknowledgments:** The authors acknowledge the GEOS-Chem model and Atmospheric Chemistry Modeling Group at Harvard University and the Atmospheric Composition Analysis Group at Washington University. We acknowledge all ozonesonde providers and their funding agencies for performing regular sonde measurements and thank the WOUDC network for archiving these data. The same applies to the teams from the lidar stations we used. We thank the OMPS Level 2 science team for providing OMPS_LP data and the MLS Level 2 science team for providing MLS data. Acknowledgement also goes to the European Space Agency for TROPOMI Level 1 data. We would also like to thank Alex Rozanov from the Department of Remote Sensing, Institute of Environmental Physics, University of Bremen, Germany, for helpful advice on using SCIATRAN.

**Conflicts of Interest:** The authors declare no conflicts of interest.

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
