# Peer review of "Ozone Profile Retrieval Algorithm Based on GEOS-Chem Model in the Middle and Upper Atmosphere"

_remotesensing, doi:10.3390/rs16081335_

Round 1

Reviewer 1 Report

Comments and Suggestions for Authors

The paper describes an improved ozone profile retrieval using GEOS-CHEM profiles as inputs. The results are compares with reference measurements and with retrieval results using profiles derived from climatology. 

General: 

  • More discussion on comparison results with respect to the the uncertainties should be provided. A more complete uncertainty discussion should show the statistical significance of the differences between the various retrievals. Some of the comments below address this issue. 

  •  
  • The manuscript should be reviewed to improve the language usage.  Some corrections are given below. 

  •  

Specific: 

54: Please reword. I think this should read something like “fewer limb payloads are available 

62: Remove “together” 

66: Remove “by”, insert “using” so that it readsusing up to 12 discrete...” 

90-91: The statement about using the profiles to monitor the coronavirus disease doesn’t fit in this discussion. Please remove or discuss elsewhere. 

100: Are the TpO3 climatology profiles related to any of the profiles that are described in this discussion. I recommend you provide context to the profiles you use to compare with GEOS-CHEM here. 

109: Please clarify what you mean byozone column concentration as an index”? 

116: Is this the first time GEOS-CHEM profiles have been used to provide a priori ozone profiles for retrieving ozone? If so, perhaps this can be emphasized a bit more. 

136 The measurements are close? 1000 km and 1.5 hours sounds like large numbers. Are these typical for these types of inter-comparisons? What is the average distance between measurement locations? 

178: Please provide a reference related to the recalibration. 

195: Change “improving to improved. 

199: Change “simulated” to “simulate. 

225: Change to “ozonesonde data are from ... 

231: It’s not clear to me what the significance of discussing the measurement principle: titration of ozone in a potassium iodide sensing solution. Is this more accurate/precise than ozonsondes based on different designs? Please add some explanation. 

242: The wording of this sentence is unclear. Please rewrite. 

Figure 5 – Can you explain why some of the profile structure (local maxima) in the reference are not captured by either GEOS of TpO3? 

440 What is an acceptable range based on the uncertainties given earlier? 

446 Eq. 7 seems to have a formatting issue. 

450 I recommend adding a histogram of the differences or plot of the differences as a function of altitude (in addition to Table 4). 

Fig. 6 Please show the 1-1 line in the plots. 

472 typo Change “pervious” to “previous 

498 What is the significance of the two ranges? Why is there a larger spread in the lower altitudes? Is this expected? 

Figure 8: The discrepancy in the peak in (d) seems larger than in the others. It looks like the simulated profiles misses a second peak. Can you explain? 

Table 5 – A histogram or plot (differences as a function of height) would be helpful here too. 

Fig. 9: Please show the 1-1 line 

Fig. 10 Why are different ranges selected here compared to the previous comparison? 

594-598: I don’t quite follow this point. Maybe I’m just confused by the expression “on the other hand.” I believe you are adding additional evidence supporting the same point made earlier in the paragraph – that the trends using the GEOS-CHEM profiles give results that match the ground results more closely than the Tp03 profiles. 

Fig 11: Why are the results plotted differently in this comparison (This type of plot would work well in the earlier part)? 

671: Are the differences statistically significant given the uncertainties in the reference measurement?

Comments on the Quality of English Language

Some corrections are given in the previous comments but the manuscript should be reviewed to improve the language usage.

Author Response

Dear Anonymous Reviewer 1,

Thank you for your hard work and constructive comments on “The ozone profile retrieval algorithm based on GEOS-Chem model in the middle and upper atmosphere”. Those comments are all valuable and very helpful for revising and improving our manuscript, as well as the important guiding significance to our research. We have studied comments carefully and considered them in the revised version of paper. Please see the attachment

Kind regards,

Yuan An

Reviewer 2 Report

Comments and Suggestions for Authors

The paper presents a nice analysis on showing that the GEOS-Chem model significantly improve the retrieval of ozone profiles form satellites measurements. The evaluation is conducted using ozone sondes and ground-based lidar measurements. The paper starts with a well-documented introduction. But the paper is too long (800 lines), same things are said several times, most of the results in the text are already in the table, and so on... I am sure that the results can be given in a more compact way (400 or 500 lines maximum). And also, the conclusion is not really a conclusion, but a long summary on the long discussion of the values.

Minor comments:

The authors said « priori ». I am not sure it the correct term, since we often use “a priori”.

The references announcement in the text do not follow the MDPI recommendations, as well as the variable line spacing.

Line128: Do the authors mean that they used laboratory spectroscopic data or “empirical” data reconstructed from TROPOM?

Table 2:  Why the name of the columns appears twice?

Lines 224-225: Could the authors  provide a refence or a justification of the criteria?

Line 229: Remove the comma.

The sentence in lines 231-232 is unnecessary.

Lines 313-316: Do the author speak in concentrations or in mixing ratios ?

Figures 3 and 4: Could the authors provide the year of the measurements?

Equation 7: It seems that the letter “nd” are missing after “grou”.

Figure 6: The R-values inside the figure are difficult to read.

Line 548: It is not so evident just when considering Figure 9.

Lines 655-656 : The sentence is unclear.

Comments on the Quality of English Language

The english should be improved, since some sentences are difficult to understand.

Author Response

Dear Anonymous Reviewer 2,

Thank you for your careful work and constructive comments on “The ozone profile retrieval algorithm based on GEOS-Chem model in the middle and upper atmosphere”. The helpful comments have substantially improved our paper a lot in the English language. We agree with points raised and modify them in the revised version of the manuscript. Please see the attachment.

Kind regards,

Yuan An

Reviewer 3 Report

Comments and Suggestions for Authors

A few suggestions to improve this manuscript:

Line 21: “R” -> “Correlation coefficient R”

Line 52-53: Please add the full name of MLS and OMPS_LP when these abbreviations first appear in the text of the manuscript.

Figures 2, 6, 9: These figures are not cleat and needs be improved to at least 300 dpi.

Eq. (5): “F(x_a)” -> “F(x_m)”, please double check the expression of this expression and corresponding explanations, besides, “x_m” is the state vector at iteration step m in this equation, not “x_i”.

Eq. (6): “y(x_m)” -> “F(x_m)”, the symbol used here needs to correspond with Eq. (3).

Figures 5, 7, 8, 10, 11: I suggest adding the legends in these figures directly to help the readers to better understand with the first sight.

Eq. (7): Please double check this equation, these are some symbols missing.

Comments on the Quality of English Language

Minor editing of English language required

Author Response

Dear Anonymous Reviewer 3,

Thank you for your patient work on “The ozone profile retrieval algorithm based on GEOS-Chem model in the middle and upper atmosphere”. The helpful comments have improved our paper a lot specifically in the English language. We agree with points raised and modify them in the revised version of the manuscript. Thanks again. Please see the attachment.

Kind regards,

Yuan An

Round 2

Reviewer 1 Report

Comments and Suggestions for Authors

Thank you for addressing all of my comments so thoroughly. The manuscript will be a nice addition to the literature. 

Reviewer 2 Report

Comments and Suggestions for Authors

The authors have well considered all my comments. So, the paper can be published as it is.